# Strength and durability of indirect protection against SARS-CoV-2 infection through vaccine and infection-acquired immunity

Sophia T. Tan [1], Isabel Rodríguez-Barraquer[2], Ada T. Kwan [3], Seth Blumberg [4,5], Hailey J. Park[1], Justine Hutchinson[6], David Leidner [7], Joseph A. Lewnard [8,9,10], David Sears [11] & Nathan C. Lo [1] ✉

Early investigation revealed a reduced risk of SARS-CoV-2 infection among social contacts of COVID-19 vaccinated individuals, referred to as indirect protection. However, indirect protection from SARS-CoV-2 infection-acquired immunity and its comparative strength and durability to vaccine-derived indirect protection in the current epidemiologic context of high levels of vaccination, prior infection, and novel variants are not well characterized. Here, we show that both vaccine-derived and infection-acquired immunity independently yield indirect protection to close social contacts with key differences in their strength and waning. Analyzing anonymized SARS-CoV-2 surveillance data from 9,625 residents in California state prisons from December 2021 to December 2022, we find that vaccine-derived indirect protection against Omicron SARS-CoV-2 infection is strongest within three months of COVID-19 vaccination [30% (95% confidence interval: 20–38%)] with subsequent modest protection. Infection-acquired immunity provides 38% (24–50%) indirect protection for 6 months after SARS-CoV-2 infection, with moderate indirect protection persisting for over one year. Variant-targeted vaccines (bivalent formulation including Omicron subvariants BA.4/BA.5) confer strong indirect protection for at least three months [40% (3–63%)]. These results demonstrate that both vaccine-derived and infection-acquired immunity can reduce SARS-CoV-2 transmission which is important for understanding long-term transmission dynamics and can guide public health intervention, especially in high-risk environments such as prisons.

Transmission dynamics of SARS-CoV-2 are driven in part by population immunity generated from vaccine-derived and infection-acquired immunity, which confer both direct and indirect protection[1,2]. Direct protection is the benefit to an individual with vaccine-derived and/or infection-acquired immunity against developing an infection or disease after exposure[2]. In contrast, indirect protection refers to the reduced risk of infection among social contacts of individuals with vaccine-derived and/or infection-acquired immunity due to infection prevention (sterilizing immunity) and reduced infectiousness[2,3]. Understanding the dynamics of vaccine-derived and infection-

acquired indirect protection from COVID-19 vaccines and prior SARS-CoV-2 infection is needed to understand population-level transmission of SARS-CoV-2 (e.g., epidemic waves due to waning population immunity), especially in settings with high population immunity, and to guide public health control measures (e.g., value of additional vaccine doses)[1].

Indirect protection against SARS-CoV-2 infection has primarily been shown through studies that found a reduced risk of transmission (or infectiousness) among COVID-19 vaccinated individuals with breakthrough SARS-CoV-2 infections[4–10], with some evidence that suggests reinfections are less infectious than primary infections[4]. Only a few studies have been able to quantify overall indirect protection, which includes indirect protection due to infection prevention in addition to reduced infectiousness[3,11]. These studies have quantified overall indirect protection in unvaccinated individuals in immunologically naïve populations, so there is limited evidence on the role of indirect protection in mitigating transmission in contemporary populations with high levels of vaccination against, and natural infection with, emerging novel variants. The temporal dynamics (strength and durability over time) of both vaccine-derived and infection-acquired indirect protection and the benefits of additional COVID-19 vaccination (booster doses) and variant-targeting vaccines (e.g., bivalent vaccine or variant-targeted monovalent formulations) also remain unclear. Investigation on these topics has been limited, in part given its study requires intensive testing of a large study population over the entire pandemic.

Studying the impact of indirect protection from vaccine-derived and infection-acquired immunity is of particular relevance to public health control measures in high-transmission environments such as prisons[4]. At the start of the COVID-19 pandemic, prison populations experienced rates of infection more than five times higher than the general population in the United States[12,13]. The incarcerated population has since experienced sustained SARS-CoV-2 transmission due to dense congregate living and suboptimal ventilation conditions[13–16]. Characterizing the dynamics of indirect protection can improve understanding of population susceptibility over time and inform public health control measures in these high-risk populations, such as optimizing the timing between additional vaccine doses to slow transmission and informing reactive vaccination efforts during outbreaks.

In this study, we use a modified test-negative case-control design to quantify overall indirect protection from both vaccine-derived and infection-acquired immunity, including their strength and durability, and considering different vaccine formulations. We perform our study retrospectively within a SARS-CoV-2 surveillance program in the California prison system, which utilized widespread testing among residents and isolation of cases to reduce transmission. The study aims to understand the complex dynamics of indirect protection for SARS-CoV-2 infection and is directly relevant to infection control and vaccine measures in the incarcerated population and other high-risk environments.

## Results

To measure vaccine-derived and infection-acquired indirect protection from COVID-19 vaccines and prior SARS-CoV-2 infection, we conducted a retrospective test-negative case-control study using anonymized data from California Correctional Health Care Services (CCHCS) and their system-wide SARS-CoV-2 surveillance program of 177,319 residents across 35 California state prisons. We defined indirect protection as the difference in SARS-CoV-2 infection risk between individuals living with roommates with and without vaccine-derived and/or infection-acquired immunity. We used a test-negative design to ensure similar testing practices between cases and controls and a clearly defined period of exposure[17,18]. The study period was from December 15, 2021, to December 15, 2022, to study the Omicron

variant/sub-variants over a period with consistent, high-volume testing within the surveillance program (Fig. 1 and Supplementary Table 1). The dominant circulating variants during this study period were Omicron BA.1, BA.2, BA.4, and BA.5 based on genomic surveillance in a subset of isolates, which also reflect variants circulating in California and the United States[19]. Descriptions of the system-wide testing, quarantine, and isolation practices implemented to reduce transmission are included in the Supplementary Notes.

### Study population

Over the study period, we identified 36,754 confirmed SARS-CoV-2 infections, 11,331 of which were reinfections (Figs. 1 and 2). In December 2021, 62% of residents had received at least one vaccine dose, and 17% of residents had received at least one booster dose (Fig. 1). Most residents received mRNA vaccines for their primary series (83%). We identified 6472 COVID-19 cases meeting our study criteria, defined as a resident with a positive SARS-CoV-2 test, residence in a two-person room, and without a positive test within the preceding 90 days. Subsequently, we identified 246,444 potential controls, defined as a resident with a negative SARS-CoV-2 test, residence in a two-person room, and without a positive SARS-CoV-2 test within the preceding 90 days or following 14 days. Residents could have multiple positive and/or negative SARS-CoV-2 tests that met study criteria (Supplementary Notes). We defined their roommate based on the housing arrangements 3–6 days prior to test collection in the case or control to reflect the biological latent period between exposure and detectable infection and movement of residents for quarantine after SARS-CoV-2 exposure. We tested alternative timings of the roommate definition in sensitivity analyses. We required that cases, controls, and their roommates were incarcerated before April 2020 to ensure complete record of prior SARS-CoV-2 infection. A complete description of inclusion and exclusion criteria can be found in the Methods and Fig. 2.

Cases and controls were matched in a variable 1:2 ratio by time (tests within two days), COVID-19 vaccine status (by dose), prior SARS-CoV-2 infection status, time since their last vaccine dose and/or infection, and building of residence and security level, which largely determined activities and number of possible social contacts, and demographic factors. Cases and controls were matched exactly by their vaccine and prior infection status to isolate differential risk due to differences in the indirect protection from their roommates' vaccine-derived and infection-acquired immunity. The final sample size included 4640 cases and 7824 controls; 3184 cases were matched to two controls and 1456 cases were matched to a single control. Match quality of cases and controls is shown in Supplementary Fig. 1. Test acceptance in the 14 days prior to study inclusion was similar between cases and controls (61% of cases and 63% of controls had a SARS-CoV-2 test) and between roommates of cases and controls (61% of case roommates and 63% of control roommates had a SARS-CoV-2 test). Roommates of cases were more likely to have a new SARS-CoV-2 infection within the preceding 4 days (44% of case roommates compared to 15% of control roommates with testing data in this period) (Supplementary Table 1). Characteristics of the cases, matched controls, and their roommates are shown in Table 1.

### Indirect protection from vaccine-derived and infection-acquired immunity

Cases were more likely to co-reside with unvaccinated individuals (14% of cases had an unvaccinated roommate compared to 12% of controls) and previously uninfected individuals (53% of cases had roommates without documented prior SARS-CoV-2 infection compared to 49% of controls).

With an adjusted model, co-residing with a vaccinated resident (received any COVID-19 vaccine dose at least 14 days prior to study inclusion) was associated with 22% indirect protection (95% CI: 13–31%)

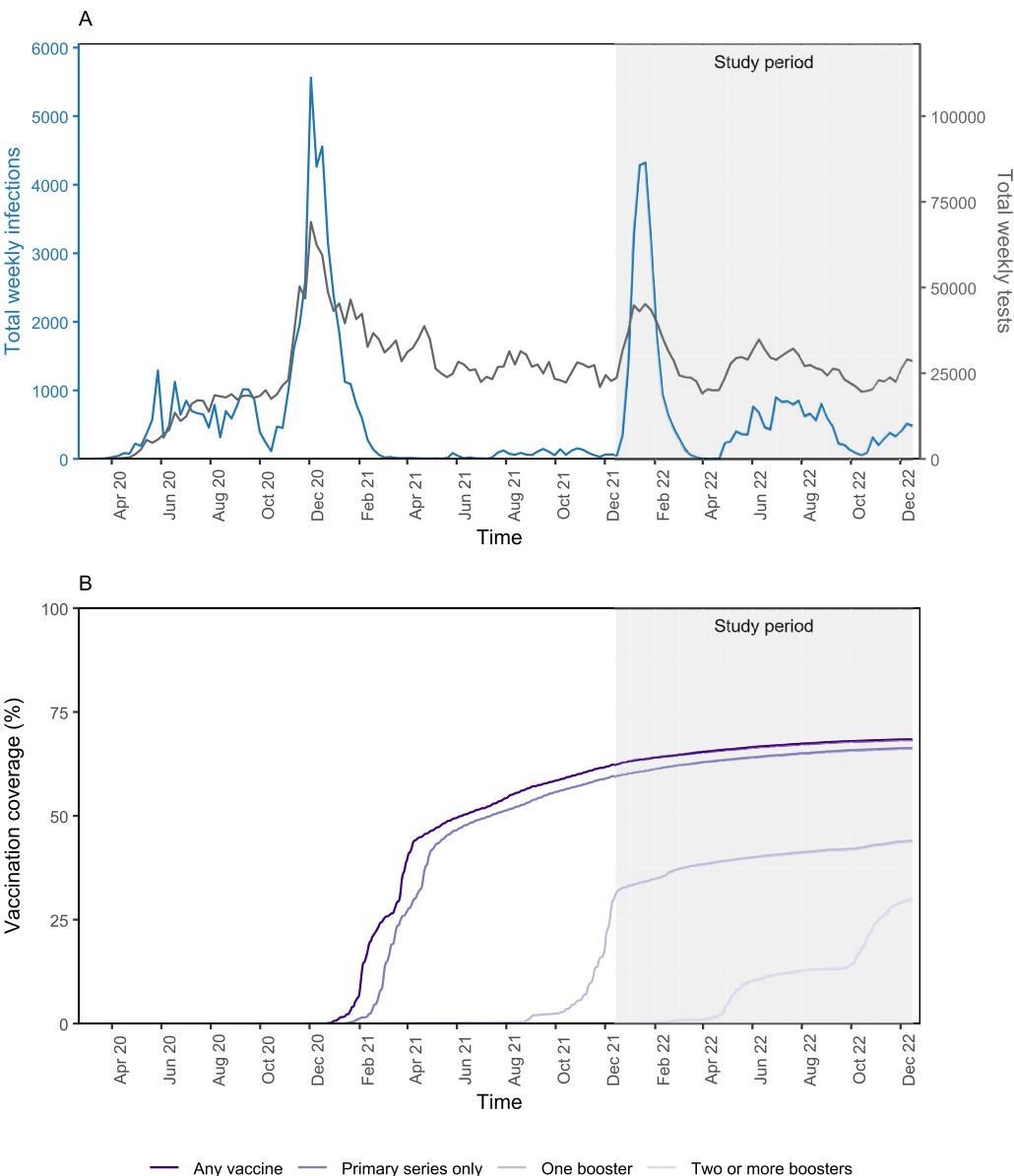

**Fig. 1 | SARS-CoV-2 infections, testing, and vaccination in California state prisons.** We analyzed anonymized retrospective data from a SARS-CoV-2 surveillance program of residents incarcerated across 35 California state prisons. We plotted the total number of weekly SARS-CoV-2 infections and weekly SARS-CoV-2 tests (top) and cumulative COVID-19 vaccine coverage (bottom) in the entire population. Data are shown from March 2020 to December 2022, although the study period was December 2021 to December 2022 (shaded in gray) during circulation of the Omicron variant/sub-variants. SARS-CoV-2 testing rates were consistent over the study period. COVID-19 vaccine administration switched from ancestral monovalent vaccines to bivalent vaccines in September 2022.

against SARS-CoV-2 infection (Fig. 3 and Supplementary Table 2). When defining vaccine status by the number of COVID-19 doses received by roommates, we found that each additional vaccine dose was associated with 7% indirect protection (95% CI: 4–11%), with up to 27% indirect protection (95% CI: 24–29%) from a resident with two or more booster doses (Fig. 3 and Supplementary Table 3). Co-residing with a resident with prior SARS-CoV-2 infection (at least 14 days prior to study inclusion) was associated with 16% indirect protection (95% CI: 8–23%) (Fig. 3 and Supplementary Table 2). We found co-residing with a resident with hybrid immunity (both vaccine-derived and infection-acquired immunity) was associated with 36% indirect protection (95% CI: 25–46%) (Fig. 3 and Supplementary Table 4). On average, roommates were more recently vaccinated (mean of 149 days prior to study inclusion) than recently infected (mean of 460 days prior to study inclusion) (Table 1).

## Strength and waning of indirect protection from vaccine-derived and infection-acquired immunity

We assessed the strength and durability of indirect protection from vaccine-derived and infection-acquired immunity over time since most recent COVID-19 vaccine, SARS-CoV-2 infection, or vaccine/infection. Co-residing with an individual who received a COVID-19 vaccine dose within three months was associated with 30% indirect protection (95% CI: 20–38%) (Fig. 4 and Supplementary Table 5), with 35% (95% CI: 23–45%) indirect protection in the first month after vaccination (Supplementary Fig. 2 and Supplementary Table 6). Indirect protection from vaccine-derived immunity demonstrated waning after 3 months, with some subsequent modest protection (indirect protection ranged from 13 to 18%) (Fig. 4 and Supplementary Table 5).

Indirect protection from infection-acquired immunity in the study population was stronger and more durable over time than indirect

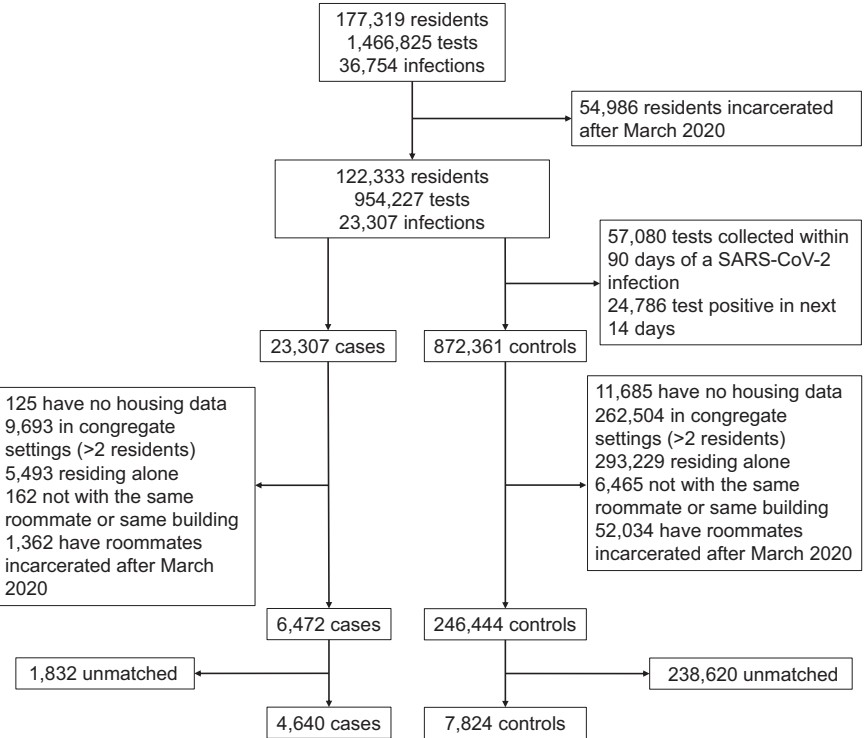

**Fig. 2 | Study population flow chart.** We designed a test-negative case-control study to measure the indirect protection provided by vaccine-derived and infection-acquired immunity from COVID-19 vaccines and prior SARS-CoV-2 infection. We analyzed anonymized retrospective data from a SARS-CoV-2 surveillance program of residents incarcerated in the California state prison system from December 15, 2021, to December 15, 2022. We identified individuals with a new SARS-CoV-2 infection (cases) and individuals with a negative SARS-CoV-2 test (controls) at the same time in the same building. Cases and controls were required to co-reside in rooms with a single other resident in the 3–6 days leading up to test collection to account for the latent period from exposure to detectable infection. We required cases and controls and their roommates to be incarcerated since March 2020 to ensure a complete record of prior infection over the pandemic. Cases and controls were matched in a 1:2 ratio based on multiple characteristics (although a subset was only matched 1:1 based on available controls meeting study criteria), including vaccination and prior infection status. We then evaluated differences in SARS-CoV-2 infection outcome in cases/controls based on the vaccine and prior infection history of their roommate. The sample size of the study population is shown at various stages of applying the study criteria and matching.

protection from vaccine-derived immunity. Co-residing with an individual with a SARS-CoV-2 infection within six months was associated with 38% indirect protection (95% CI: 24–50%). Indirect protection from infection-acquired immunity persisted over one year; prior SARS-CoV-2 infection more than one year ago was associated with 12% indirect protection (95% CI: 3–19%) (Fig. 4 and Supplementary Table 5). When measuring time since either the most recent vaccination or infection, indirect protection demonstrated waning over time (Fig. 4 and Supplementary Table 5).

We did not detect a multiplicative interaction between vaccine-derived and infection-acquired indirect protection, which suggests the strength of vaccine-derived indirect protection from COVID-19 vaccines does not differ between roommates with and without evidence of prior SARS-CoV-2 infection (Supplementary Table 9). Additionally, both vaccine-derived and infection-acquired indirect protection from roommates were broadly similar among cases and controls with different levels of direct protection (no prior immunity, vaccine-derived immunity only, infection-acquired immunity only, and hybrid immunity), although this analysis was underpowered in some groups (Supplementary Table 10).

### Indirect protection from variant-targeted COVID-19 vaccines
Starting September 2022, residents received bivalent vaccine doses, which targeted both the ancestral SARS-CoV-2 strain and Omicron variants BA.4 and BA.5 to match circulating Omicron subvariants[20]. By December 2022, 20.5% of residents received a bivalent vaccine dose. We found that a bivalent COVID-19 vaccine dose provided 40% indirect protection (95% CI: 3–63%) within the first three months of vaccine receipt (September – December 2022; Fig. 5 and Supplementary Table 11; Supplementary Notes).

### Negative control analysis
As a negative control analysis, we tested a negative control exposure of influenza vaccination in roommates. We found no evidence of indirect protection against SARS-CoV-2 infection from influenza vaccination within the past year (0.6% [95% CI: (-8–8%)]) (Supplementary Table 12).

### Sensitivity analyses
Our findings were similar with different definitions for timing of co-residence for cases and controls and their roommates. When defining co-residence on the third day prior to test collection in cases and controls, we found having a roommate with vaccine-derived or infection-acquired immunity was associated with 23% (95% CI: 14–31%) and 16% (95% CI: 9–22%) indirect protection, respectively. When defining co-residence as the week leading up to test collection, we estimated indirect protection was 22% (95% CI: 13–31%) from vaccine-derived immunity and 12% (4–19%) from infection-acquired immunity (Supplementary Table 13). Study findings were robust to alternative specifications in matching and statistical model, including 1:1 matching, alternative matching criteria, and alternative model covariates (Supplementary Table 14). We found similar results when we used an unconditional logistic regression, controlling for all matched factors, and when we accounted for repeated observations of residents in the study population over time (Supplementary Table 15).

**Table 1 | Characteristics of the study population including COVID-19 cases, matched controls, and their roommates in California prisons**

| | Outcome (N (SD) or N (%)) | | | Exposure (N (SD) or N (%)) | | |
|---|---|---|---|---|---|---|
| | Cases (N = 4,640) | Controls[a] | | Roommate of cases (N = 4,640) | Roommate of controls[a] | |
| | | Unweighted (N = 7,824) | Weighted (N = 4,640) | | Unweighted (N = 7,824) | Weighted (N = 4,640) |
| **Age (years)** | 42.3 (12.6) | 42.6 (12) | 42.3 (11.9) | 42.7 (12.4) | 42.4 (11.9) | 42.2 (11.9) |
| **Sex (male)** | 4,519 (97%) | 7,616 (97%) | 4,518 (97%) | 4,517 (97%) | 7,617 (97%) | 4,518 (97%) |
| **Race/ethnicity** | | | | | | |
| American Indian/Alaskan Native | 52 (1%) | 87 (1%) | 51 (1%) | 52 (1%) | 78 (1%) | 47 (1%) |
| Asian or Pacific Islander | 52 (1%) | 118 (2%) | 71 (2%) | 49 (1%) | 92 (1%) | 54 (1%) |
| Black | 1,285 (28%) | 2,084 (27%) | 1,258 (27%) | 1,272 (27%) | 2,088 (27%) | 1,258 (27%) |
| Hispanic | 2,321 (50%) | 3,988 (51%) | 2,346 (51%) | 2,324 (50%) | 4,016 (51%) | 2,374 (51%) |
| White | 738 (16%) | 1,179 (15%) | 704 (15%) | 733 (16%) | 1,197 (15%) | 700 (15%) |
| Other | 192 (4%) | 368 (5%) | 210 (5%) | 210 (5%) | 353 (5%) | 208 (4%) |
| **Risk score for severe COVID-19[b]** | 1.4 (2) | 1.3 (1.8) | 1.3 (1.8) | 1.4 (2) | 1.3 (1.8) | 1.3 (1.8) |
| **Security level[c]** | | | | | | |
| Low | 2,218 (48%) | 3,907 (50%) | 2,218 (48%) | 2,218 (48%) | 3,907 (50%) | 2,218 (48%) |
| Moderate | 899 (19%) | 1,455 (18%) | 899 (19%) | 899 (19%) | 1,455 (18%) | 899 (19%) |
| High | 1,523 (33%) | 2,553 (32%) | 1,523 (33%) | 1,523 (33%) | 2,553 (32%) | 1,523 (33%) |
| **COVID-19 vaccine status** | | | | | | |
| Unvaccinated | 541 (12%) | 817 (10%) | 541 (12%) | 652 (14%) | 897 (11%) | 545 (12%) |
| Partially vaccinated | 22 (0%) | 27 (0%) | 22 (0%) | 48 (1%) | 113 (1%) | 70 (2%) |
| Primary series only | 960 (21%) | 1,503 (19%) | 960 (21%) | 979 (21%) | 1,648 (21%) | 1,004 (22%) |
| 1 booster dose | 2,773 (60%) | 4,881 (62%) | 2,773 (60%) | 2,579 (56%) | 4,533 (58%) | 2,636 (57%) |
| 2+ booster doses | 344 (7%) | 596 (8%) | 344 (7%) | 382 (8%) | 633 (8%) | 385 (8%) |
| Time since last vaccine dose (days) | 155 (122) | 147 (120) | 150 (122) | 151 (125) | 148 (128) | 149 (129) |
| **Prior SARS-CoV-2 infection status** | | | | | | |
| Has prior infection | 1,953 (42%) | 3,471 (44%) | 1,953 (42%) | 2,181 (47%) | 4,024 (51%) | 2,344 (51%) |
| Time since last infection (days) | 486 (147) | 473 (152) | 469 (153) | 469 (169) | 455 (175) | 453 (174) |

In this analysis, we estimated indirect protection against SARS-CoV-2 infection from differences in vaccine-derived immunity (from COVID-19 vaccines) and infection-acquired immunity (from prior SARS-CoV-2 infection) generated by roommates of matched cases and controls. Cases were matched with controls in a 1:2 ratio, although a subset were matched 1:1 based on available controls meeting study criteria. Cases and controls were matched exactly by vaccine status, prior infection status, building, and security level, with additional distance matching on other variables. Full matching process is described in the Methods.
[a]Differences in estimates for variables matched exactly between cases and unweighted controls (i.e., vaccine status, prior infection status, and security level) are due to variable 1:2 matching. We report weighted estimates of these variables to account for differences in match group size.
[b]Risk for severe COVID-19 is a composite score of the number of risk factors for severe disease including age and medical conditions. Risk scores were calculated by the California Correctional Health Care Services.
[c]Resident contact and activity are determined by their security level. Low level (score 1-2) reflects the lowest security level with more social contacts, while high level (score 4) is the highest security level with fewer social contacts.

## Discussion

In this study, we found that both vaccine-derived and infection-acquired immunity yield measurable and meaningful indirect protection against Omicron SARS-CoV-2 infection, indicating their role in reducing transmission. Our results suggest that infection-acquired immunity from prior SARS-CoV-2 infection provides stronger and more durable indirect protection than vaccine-derived immunity from COVID-19 vaccines, and indirect protection from both wanes over time. We also evaluated indirect protection from variant-targeting (bivalent) vaccines and found they provide strong indirect protection within the first three months of vaccination. Our findings have implications for understanding long-term viral transmission dynamics of SARS-CoV-2, which are likely governed by these indirect effects (i.e. rise in transmission and epidemic waves due in part to waning indirect protection). Our results also highlight that indirect protection from vaccines exist in a population with high cumulative vaccination and prior infection during a period of intense transmission, which is most relevant to this stage of the COVID-19 pandemic. This work has relevance to vaccine policy and public health measures in high-risk environments, such as reactive vaccination during periods of

outbreaks and/or consideration of higher frequency of routine vaccination to maximize indirect protection.

We observe infection-acquired immunity may generate stronger and more durable indirect protection than vaccine-derived immunity. This finding becomes most clear when accounting for time since vaccine and/or infection because the study population is more recently vaccinated than infected. The strength of infection-acquired indirect protection could be explained by many mechanisms, including that natural infection generates a more robust immune response. Another explanation is that recent infection occurs with more contemporary variants compared to vaccination with an ancestral strain formulation, meaning infection generates an immune response more tailored to circulating variants. We also find that indirect protection from both vaccine-derived and infection-acquired immunity wanes over time. This could be explained by many immunologic mechanisms (e.g., waning antibody titers), as well as by viral evolution and increasing mismatch between the generated host immunity and circulating variants over time.

Additional considerations strengthen the study findings. We identify strong and consistent indirect protection generated by

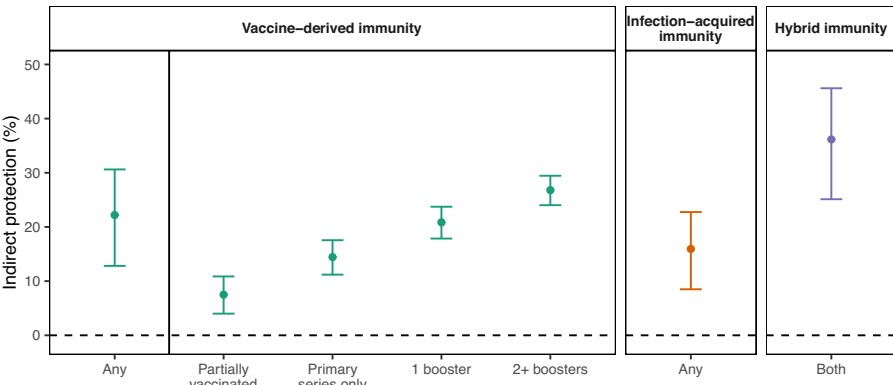

**Fig. 3 | Overall vaccine-derived and infection-acquired indirect protection to close social contacts against SARS-CoV-2 infection.** We estimated the indirect protection that vaccine-derived and infection-acquired immunity provided to a roommate. We defined indirect protection as the change in risk of SARS-CoV-2 infection in an individual based on their roommate's COVID-19 vaccine and prior SARS-CoV-2 infection status. We adjusted for age and risk of severe COVID-19 of both the case and control and their roommates. Residents in California state prisons were less likely to be infected by the Omicron SARS-CoV-2 variant if they co-resided with an individual with vaccine-derived and/or infection-acquired immunity. The mechanism of protection is likely that individuals with vaccination and/or prior infection are less likely to become infected (and then transmit infection) or are less infectious upon breakthrough infection or reinfection. Residents with hybrid immunity were more likely to be recently vaccinated than recently infected. The average time since last vaccine dose was more recent than average time since last infection. We plotted the mean (point estimate) and associated 95% confidence intervals (bars) for indirect protection from vaccine-derived immunity (green), infection-acquired immunity (orange), and hybrid immunity (purple). Separate regression models were fit for any vaccine and any infection (Supplementary Table 2), vaccination by dose (Supplementary Table 3), and hybrid immunity (Supplementary Table 4).

vaccine-derived and infection-acquired immunity across multiple primary and sensitivity analyses, accounting for both mechanisms of indirect protection: infection prevention and reduced infectiousness upon infection. We find vaccine-derived indirect protection is dose-dependent, and indirect protection from both vaccine-derived and infection-acquired immunity are time-dependent. The dose- and time-dependence are not only scientifically and policy relevant, but also strengthen the validity of our findings and reduce the likelihood of confounding. The time-dependence in indirect protection found in our study also matches literature on the comparative strength and waning of direct protection from vaccine-derived and infection-acquired immunity[21–23]. We find that roommates of controls, who were more likely to have a history of prior vaccination and SARS-CoV-2 infection, were also less likely to have a positive SARS-CoV-2 test in the 4 days prior to study inclusion than roommates of cases (Supplementary Table 1). This evidence supports the mechanism of indirect protection, e.g., reduction of infection risk in the roommate due to vaccination and/or infection-acquired immunity. Finally, we employ a negative control exposure (influenza vaccination) and found no evidence for indirect protection against SARS-CoV-2 infection from influenza vaccination, which strengthens the causal implications of vaccine-derived indirect protection from COVID-19 vaccines in our study population.

This study is important for improving health and addressing inequities in the incarcerated population due to COVID-19. Over the pandemic, this population has experienced high rates of SARS-CoV-2 transmission largely due to structural and environmental risk factors, including dense congregate living and suboptimal ventilation conditions[12–16]. Although severe COVID-19 (defined as hospitalization or death) outcomes were rare during the Omicron era in this population, risk for complications (such as long COVID) and high SARS-CoV-2 infection incidence, despite high levels of vaccine coverage and prior infection, highlight the need for continued optimization of health policy in this vulnerable population. We find evidence that additional vaccine doses provide additive indirect protection to close social contacts, even in individuals with infection-acquired immunity, which suggests that residents who co-reside together can provide and receive indirect benefits that reduce transmission risk with additional vaccination. Housing arrangements based on vaccine and prior infection status may be particularly relevant for high-risk residents (e.g., 65+ years, immunocompromised, immunologically naive) and may reduce their relative risk of infection by nearly 40%. Furthermore, we find transmission often occurs outside of their rooms (many cases were not exposed to a SARS-CoV-2 positive roommate leading up to test collection) (Supplementary Table 1) and benefits of indirect protection could extend to a broader spatial scale (e.g., building level). Therefore, increasing uptake of additional vaccine doses in residents and especially among staff, where even vaccination with primary series alone has lagged behind residents (65%)[24], would likely reduce transmission. Our findings also suggest that reactive vaccination campaigns could be used to mitigate transmission (via vaccine-derived indirect effects on transmission) during future outbreaks, especially if future variants are more virulent, and residents may benefit from considering more frequent routine COVID-19 vaccination. This benefit should be balanced with the risk of myocarditis related to mRNA COVID-19 vaccines[25].

This study has limitations. This study design does not distinguish between indirect protection from infection prevention and indirect protection from reduced infectiousness; the relative contribution of these mechanisms may vary between individuals (e.g., immunity status). This test-negative case-control study design is observational and thus subject to potential confounding[26]. For example, residents with vaccine-derived and/or infection-acquired immunity may have unobserved differences in behavior (e.g., fewer social interactions, a more risk-averse social cohort, masking) that lower their risk of infection and/or transmission to social contacts, and we do not have data on behavior such as masking. However, our analysis demonstrates time-dependent indirect protection (waning) from both sources, which argues against the presence of such confounding factors. The negative control analysis (influenza vaccine exposure) further reduces the likelihood of residual confounding. We also match cases and controls by all observed characteristics (including vaccine and prior infection status), and roommate assignments are expected to be random with respect to vaccine and prior infection status. While testing is frequent in this population throughout the pandemic, there may be some misclassification of prior SARS-CoV-2 infection status or timing due to imperfect case ascertainment[26], though any misclassification is more likely nondifferential (Supplementary Table 1). While cases are required to have a new SARS-CoV-2 infection, timing of exposure and

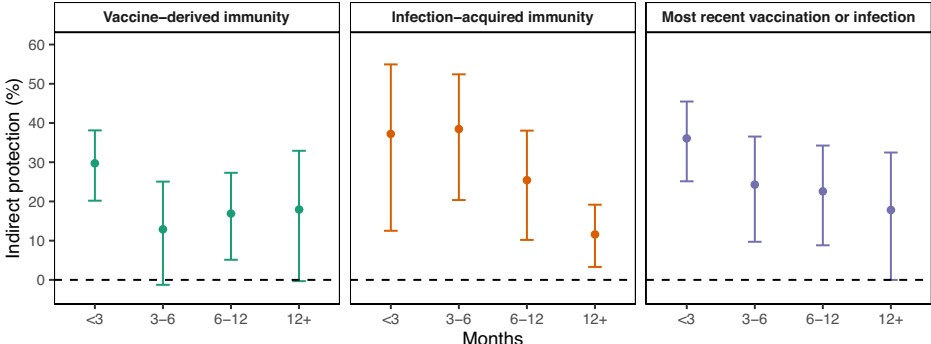

**Fig. 4 | Comparative strength and durability of vaccine-derived and infection-acquired indirect protection to close social contacts against SARS-CoV-2 infection.** We estimated the strength and durability of indirect protection that COVID-19 vaccine-derived and SARS-CoV-2 infection-acquired immunity provided to a close social contact (roommate). Residents in California state prisons were less likely to be infected by the Omicron SARS-CoV-2 variant when residing with an individual with infection-acquired or vaccine-derived immunity; both sources of indirect protection waned over time, but infection-acquired immunity yielded stronger and more durable indirect protection. We estimated indirect protection based on time since last vaccine dose (green), time since last SARS-CoV-2 infection (orange), and time since most recent vaccine or infection (purple). We plotted the mean (point estimate) and associated 95% confidence intervals (bars) for indirect protection. We fit separate models for vaccine-derived immunity, infection-acquired immunity, and most recent vaccine or infection (Supplementary Table 5).

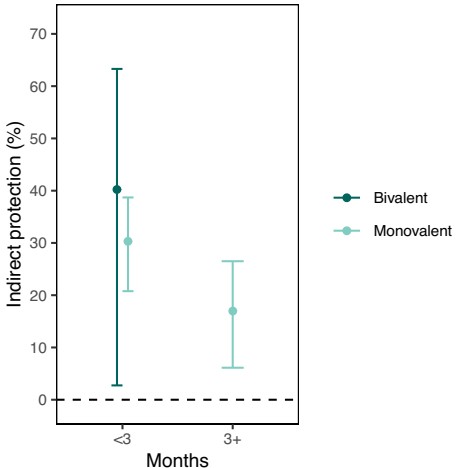

**Fig. 5 | Indirect protection from variant-targeting bivalent COVID-19 vaccine and comparison to ancestral monovalent vaccine during Omicron era.** We estimated the indirect protection from a variant-targeting vaccine (bivalent vaccine formulated with ancestral strain and Omicron subvariants BA.4/5 to target circulating variants) in California state prisons. Our goal was to determine how concordance of COVID-19 vaccine formulation and circulating SARS-CoV-2 variants affects vaccine-derived indirect protection. Administration of bivalent vaccines began in September 2022, which yielded approximately three months of follow up; therefore, estimates were only available for three months. We include estimates of indirect protection from ancestral monovalent vaccines during the study period prior to bivalent vaccine introduction. The average time since bivalent vaccine was more recent than average time since last monovalent vaccine (within three months; Supplementary Table 11). We estimated indirect protection and plotted the mean (point estimate) and associated 95% confidence intervals (bars) for indirect protection.

infectiousness are unknown. However, we observed robust results when we varied our definition for timing of co-residence and roommate definitions. We also did not have access to cycle threshold values for PCR tests or any individual-level data on symptoms, serologic status, or viral genome for cases. Our analysis examines indirect protection at the room level to focus on the benefit of room-level housing arrangement policies that consider vaccine or prior infection status, although a substantial fraction of SARS-CoV-2 transmission occurs outside the room through interactions with other residents and staff. A higher proportion of outside-room transmission would reduce the indirect protection conferred by a roommate, so our analysis may underestimate the population-level indirect protection generated by vaccine-derived and infection-acquired immunity. The absolute magnitude of indirect protection against SARS-CoV-2 infection will depend on the transmission environment and social context. This study focuses on indirect protection from Omicron SARS-CoV-2 infection, though these findings may generalize to other variants and vaccine formulations. Our study population is a subset of the overall California incarcerated population during the pandemic that co-resides in close contact with a single person, and our specific estimates on indirect protection are most applicable to similar high-risk transmission environments.

This study finds that vaccine-derived and infection-acquired immunity from COVID-19 vaccines and prior SARS-CoV-2 infection confer indirect protection to close social contacts, with stronger and more durable indirect protection from infection-acquired immunity. These findings have implications for understanding transmission dynamics of SARS-CoV-2 and can inform vaccine and public health control measures for high-risk environments.

## Methods
### Study design
We used a modified test-negative case-control design to evaluate indirect protection from COVID-19 vaccines and immunity from prior SARS-CoV-2 infection among residents in California prisons[17,18]. We defined indirect protection as the difference in risk of SARS-CoV-2 infection between individuals with an unvaccinated and previously uninfected roommate and individuals with a vaccinated and/or previously infected roommate. While the test-negative case-control design is most often applied to measure direct vaccine effectiveness by comparing vaccine status in cases and controls[18], we match cases and controls by COVID-19 vaccine status and prior SARS-CoV-2 infection history to identify differential risk between cases and controls solely from their roommates' vaccine-derive and infection-acquired immunity.

### Data
We used anonymized person-level data from CCHCS on demographics, SARS-CoV-2 testing, COVID-19 vaccination, and nightly housing information for residents incarcerated in the California state prison system from March 1, 2020, to December 15, 2022. We defined a study period from December 15, 2021, to December 15, 2022, based on

a pre-specified objective of studying contemporary variants (e.g., Omicron subvariants BA.1, BA.2, BA.4, and BA.5) while ensuring consistent system-wide testing practices. The majority of tests were polymerase chain reaction (78%). Residents were isolated if they tested positive for SARS-CoV-2. Isolation, quarantine, and testing practices and vaccine administration during the study period are further described in the Supplementary Notes.

## COVID-19 cases and controls

The inclusion and exclusion criteria for cases and controls are shown in Fig. 2. Cases were defined as residents with a positive SARS-CoV-2 test (first positive test in at least 90 days). Controls were defined as residents with a negative SARS-CoV-2 test with no positive test in the preceding 90 days or following 14 days. Both cases and controls were required to reside in rooms of only two residents in the 3–6 days leading up to their test. The timing of this housing requirement was chosen to represent the latent period between exposure and detectable infection and to account for movement of residents in response to SARS-CoV-2 exposure and was varied in sensitivity analyses. Cases, controls, and their roommates were required to have been incarcerated before April 1, 2020, to ensure a more complete history of documented SARS-CoV-2 infection.

## Statistical analysis

We performed matching of cases and controls by person-level characteristics and time to improve precision and control for unobserved factors (Supplementary Notes). Cases and controls were first matched exactly by time (tests within two days), building and security level (which largely determines a resident's social contacts), COVID-19 vaccine status (unvaccinated, partially vaccinated, primary series alone, one booster dose, two or more booster doses), and prior SARS-CoV-2 infection (binary). We matched cases and controls by vaccine and prior infection status to limit confounding from direct protection. Cases and controls were then further matched to minimize differences in time since their most recent SARS-CoV-2 infection and/or COVID-19 vaccine and age (in years) and risk score for severe COVID-19 (weighted score of risk factors for severe COVID-19 used by CCHCS) between cases and controls and between their roommates. Cases and controls were ineligible for matching if they resided together. Cases and controls were matched in a 1:2 ratio (although a subset was matched 1:1 based on eligibility of controls for matching) though this was varied in a sensitivity analysis. We did not allow residents to be repeated within a matched group. Descriptive data on the quality of matches are in Supplementary Fig. 1.

To estimate the indirect protection from vaccine-derived and infection-acquired immunity, we fit conditional logistic regression models, defining strata for matched sets of cases and controls[27]. The model outcome was SARS-CoV-2 infection (case or control), and the primary exposures of interest were vaccine-derived immunity from COVID-19 vaccines (binary) and infection-acquired immunity from prior SARS-CoV-2 infection (binary) in the roommate. Indirect protection (protection generated by roommate against SARS-CoV-2 infection) was estimated from one minus the adjusted odds ratio (OR)[17,18]. We adjusted for age and severe COVID-19 risk in cases/controls and their roommates. Since the model was stratified by matched case and control groups, we did not adjust for covariates that were exactly matched (e.g., building and security level and vaccine and prior infection status of cases and controls). We treated repeated observations of a single resident in different matched groups independently, although we tested this assumption in sensitivity analyses. We defined onset of vaccine protection (vaccine-derived immunity) as 14 days after receipt of each COVID-19 vaccine dose, which is consistent with literature[28,29]. We defined infection-acquired immunity as prior infection more than 14 days before study inclusion to ensure we did not include active SARS-CoV-2 infection.

We fit additional models to explore the dose- and time-dependence of indirect protection. We defined vaccine-derived immunity in roommates numerically by doses (unvaccinated, partially vaccinated, primary series alone, one booster dose, two or more booster doses). We assessed durability of indirect protection from vaccine-derived and infection-acquired immunity in separate models by defining exposures by time. We defined key time categories based on vaccine literature[23,30], including <3 months, 3–6 months, 6–12 months, and 12+ months. We also measured indirect protection over time since most recent immunizing event (most recent vaccine dose or infection). Each model was adjusted for age and risk of severe COVID-19 of cases/controls and their roommates and, if applicable, vaccine or prior infection status (binary) of roommates. We tested interactions between vaccine and prior infection exposures and measured indirect protection from hybrid immunity (both vaccine-derived and infection-acquired immunity). We stratified indirect protection by immune status in cases and controls to explore the relationship between indirect and direct protection.

Bivalent COVID-19 vaccines targeting both the ancestral SARS-CoV-2 strain and Omicron subvariants BA.4 and BA.5 became the only available vaccine in the study population in September 2022. We performed an analysis where we estimated the indirect protection from bivalent vaccines (received on or after September 1, 2022). We compared these estimates to the indirect protection from monovalent vaccine (< 3 months, 3+ months), although these estimates were from the period prior to bivalent introduction (Supplementary Notes).

We performed a negative control analysis by testing a negative control exposure of influenza vaccination in roommates (instead of COVID-19 vaccination)[31]. Influenza vaccination was chosen since it should have no causal impact on SARS-CoV-2 infection or transmission. Analysis was conducted in R (version 4.3.1). All code is publicly available[32].

## Sensitivity analyses

We conducted various sensitivity analyses of study design and analytical decisions. We tested different housing requirements for roommates of cases and controls (3 day, 0–6 days, 6–9 days). We assessed differences in results when matching cases and controls in a 1:1 ratio. We assessed robustness of results to adjustment for time since last COVID-19 vaccine dose and time since last SARS-CoV-2 infection. Since matching may introduce bias[27], we also tested an unconditional logistic regression model with the same study population of matched cases and controls and adjusted for all factors that had been matched exactly, including building and security level, vaccine status of cases and controls, and prior infection status of cases and controls. To assess the sensitivity of our results to repeated measures of the same residents, we fit a conditional logistic regression model without repeated measures and an unconditional logistic regression model with person-level cluster robust errors.

## Ethics

This study was approved by the Institutional Review Board (IRB) at Stanford University and UCSF. The IRB included a waiver of consent given use of retrospective secondary data without direct identifiers that were collected for public health surveillance. Additionally, this study was deemed to be minimal risk and has direct relevance to improving the health of the population (Supplementary Notes). Isolation, quarantine, and testing practices are further described in the Supplementary Notes.

## Reporting summary

Further information on research design is available in the Nature Portfolio Reporting Summary linked to this article.

## Data availability

Data requests may be made to the California Correctional Health Care Services and California Department of Corrections and Rehabilitation (Data.Requests@cdcr.ca.gov) and are subject to controlled access to protect privacy of residents[33]. Data access will require submission of a formal study protocol with review by the California Correctional Health Care Services. If approved, a data use agreement will be needed before data access.

## Code availability

All analytic code is publicly available[32].

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

## Acknowledgements

We thank Dr. Heidi Bauer and Dr. Richard Sun for valuable input and acknowledge individuals at the Office of the California Prison Health Care Receivership and California Correctional Health Care Services. We acknowledge the individuals who provided the data underlying these analyses. The study content is solely the responsibility of the authors and does not necessarily represent the official views of the National Institutes of Health. N.C.L. is supported by the National Institutes of Health, NIAID New Innovator Award (DP2AI170485).

## Author contributions

S.T.T. and N.C.L. had full access to all the data in the study and take responsibility for the integrity of the data and the accuracy of the data analysis. Study concept and design: S.T.T., D.S., and N.C.L.

Statistical analysis: S.T.T. and N.C.L. Analytic coding: S.T.T. and H.J.P. Acquisition, analysis, or interpretation of data: All authors. First draft of the manuscript: S.T.T. and N.C.L. Critical revision of the manuscript: All authors. Contributed intellectual material and approved final draft: All authors.

## Competing interests

J.A.L. has received grants, honoraria, and speaker fees from Pfizer; grants and honoraria from Merck, Sharp, & Dohme; honoraria from Valneva; and honoraria from VaxCyte; all unrelated to the subject of this work. A.T.K. and D.S. received funding from the California Prison Health Care Receivership. N.C.L. reports consulting fees from the World Health Organization related to guidelines on neglected tropical diseases, which are outside the scope of the present work. The remaining authors have no disclosures.

## Additional information

[1]Division of Infectious Diseases and Geographic Medicine, Department of Medicine, Stanford University, Stanford, USA. [2]Division of HIV, Infectious Diseases, and Global Medicine, Department of Medicine, University of California, San Francisco, CA, USA. [3]Division of Pulmonary and Critical Care Medicine, Department of Medicine, University of California, San Francisco, CA, USA. [4]F.I. Proctor Foundation, University of California, San Francisco, CA, USA. [5]Department of Medicine, University of California, San Francisco, CA, USA. [6]California Correctional Health Care Services, Elk Grove, CA, USA. [7]California Department of Corrections and Rehabilitation, Elk Grove, CA, USA. [8]Division of Epidemiology and Biostatistics, School of Public Health, University of California, Berkeley, CA, USA. [9]Division of Infectious Diseases and Vaccinology, School of Public Health, University of California, Berkeley, CA, USA. [10]Center for Computational Biology, College of Computing, Data Science, and Society, University of California, Berkeley, CA, USA. [11]Division of Infectious Diseases, Department of Medicine, University of California, San Francisco, CA, USA. ✉e-mail: Nathan.Lo@stanford.edu

10
