## [Transparent Peer Review file · Nature Communications]

Strength and durability of indirect protection against SARS-CoV-2 infection through vaccine and infection-acquired immunity

Corresponding Author: Dr Nathan Lo

Version 0:

Reviewer comments:

Reviewer #1

(Remarks to the Author)

Tan and colleagues estimate the indirect effect of Covid-19 vaccination and infection using data from the California state prison system. Overall, I find the study interesting and well-conducted, though I do have several comments and suggestions.

The authors correctly mention that indirect protection has two main pathways (in this context - direct effectiveness in the roommate and reduced infectiousness once the roommate is infected), mention that "few studies have been able to quantify indirect protection while accounting for both pathways of indirect protection", and even state that this study was done "... while accounting for both pathways of indirect protection (infection prevention and reduced infectiousness upon infection". But in fact, these two mechanisms were not formally modeled in this study. The authors should either formally model both parts of the effect (i.e., separately model direct effectiveness and the secondary attack rate) or acknowledge throughout the manuscript that the observed effects are a composite of two effects, and that their relative contribution might differ between individuals (e.g., might be different between previously vaccinated and previously infected).

Conclusions from the analysis of bivalent vs. monovalent vaccines (Figure 5 and text) are overstated. First, the two exposures were not directly contrasted (for relative effectiveness), rather both were compared separately to the unvaccinated. Second, estimates of 18-70% vs. 19-38% are at most "unclear" and certainly do not demonstrate any "trends" (more generally, statistically un-rigorous mentions of "trends" should be removed).

The authors do well to adjust for the immune status of the case/control. But the immune status of the case/control is scientifically very interesting as a possible effect modifier, and I would be eager to see an analysis of the indirect effect stratified on this immune status. I.e., it is plausible that the indirect effect is much smaller (perhaps entirely non-existent) when the index case is vaccinated. This would also be very relevant to public health policy.

The authors state "We performed matching of cases and controls to minimize risk of confounding" – but matching in a case-control study does not mitigate confounding (the opposite is true, as usually cited from <https://www.bmj.com/content/352/bmj.i969>, which is also cited in this manuscript). The sentence should be corrected.

Are the estimates in Figure 3 crude or adjusted? This should be written in the legend.

Perhaps I missed it, but I couldn't find which vaccines were used (other than being monovalent or bivalent) in the study population. mRNA or non-mRNA? Pfizer? Moderna? Etc.

The test-negative design mitigates certain biases while potentially causing new ones

(<https://pubmed.ncbi.nlm.nih.gov/27587721/>). This should be acknowledged, and the relevant biases discussed.

The limitation to external generalizability resulting from using data from a prison system should be acknowledged. E.g., it is plausible that the indirect effect is stronger because of the closer cohabitation that occurs in a prison.

I would alter the tone of the Discussion section to be less self-congratulatory ("our study makes significant strides", etc.). A modest discussion of each relevant point would be preferable to an enumeration of strengths.

Reviewer #2

(Remarks to the Author)

The authors take advantage of SARS-CoV-2 test data from a surveillance program in California state prisons to determine indirect protection from infection-acquired immunity and its comparative strength and durability to vaccine-derived indirect protection. The study shows that both infection-acquired and vaccine-derived immunity independently yield indirect protection to close social contacts (roommates in a prison) with differences in their strength and waning.

As the authors point out, there is in the current context of high levels of vaccination, prior infection and novel variants of SARS-CoV-2 limited information on overall indirect protection (i.e., the sum of infection prevention and reduced infectiousness upon infection). The value of the study is the fact that this was determined in a well-defined setting using a straight forward and elegant case-control design. A measurement of indirect protection from infection as well as vaccines (including ancestral and variant targeted vaccines) in one study and in a well-defined setting is an original finding, at least to my knowledge.

The study found that infection-acquired immunity generated stronger and more durable indirect protection than vaccine-derived indirect protection. This observation was expected, but the merits of the study is that this was quantified, and the temporal dynamics was explored. On this basis, I expect the findings to be of significance to the field.

The study was designed as a test-negative case-control study. Cases were residents with a positive SARS-CoV-2 test, whereas controls were residents with a negative SARS-CoV-2 test. Cases and controls were matched exactly by time (tests within two days), prison building and security level, COVID-19 vaccine status and prior SARS-CoV-2 infection. Cases and controls were then further matched to minimize differences in time since their most recent SARS-CoV-2 infection and/or COVID-19 vaccination as well as age and risk score for severe COVID-19. The analytical strategy and the validity of the results depends on the success of this complicated matching procedure, and the authors are well aware of this. This is reflected among others in the supplemental material which provides insight into the process of matching and also the sensitivity analyses presented in Suppl Table 11. Nonetheless, Table 1 shows minor differences between cases and controls, e.g., 11.5% cases unvaccinated compared with 10.5% controls, and 42% cases with prior infection compared with 44% controls. These differences may be due to the fact that some cases had two controls and others only one, and may not represent a mismatch problem. An explanation will be relevant.

The authors mention that 2179 residents were included more than once, but the observations were treated as independent. I'm not a statistician, but I would think that repeated observations should be taken into account, at least by providing robust confidence intervals (the intervals presented may be too narrow, but the estimates are likely to be valid). Most software packages will provide solutions for this. This issue should be addressed before the paper can be accepted.

A minor issue is the headings of Table 1. Many readers will look at this table before reading the methods section. The roommates reflect the exposure status of cases/controls. The table may be misunderstood in a way that the roommates represent a second set of cases/controls. Perhaps a slight edition will make it easier for the uninitiated reader to understand the design and the table.

The supplementary material presents the results of a logistic regression analysis (column 6 in Suppl table 11). It is unconditional I assume - should perhaps be spelled out.

Version 1:

Reviewer comments:

Reviewer #1

(Remarks to the Author)

The authors have responded to my comments in a satisfactory manner.

Reviewer #2

(Remarks to the Author)

Comments from both reviewers have been addressed during the revision and I have no further remarks.

Tan *et al.* “Strength and durability of indirect protection against SARS-CoV-2 infection through vaccine and infection-acquired immunity”
(Reference no. NCOMMS-24-48029-T) – Point-by-point response

Response to Reviewer 1:

Remarks to the Author

Tan and colleagues estimate the indirect effect of Covid-19 vaccination and infection using data from the California state prison system. Overall, I find the study interesting and well-conducted, though I do have several comments and suggestions.

Response: We appreciate the reviewer’s thoughtful and helpful comments.

Comment 1

The authors correctly mention that indirect protection has two main pathways (in this context - direct effectiveness in the roommate and reduced infectiousness once the roommate is infected), mention that "few studies have been able to quantify indirect protection while accounting for both pathways of indirect protection", and even state that this study was done "...while accounting for both pathways of indirect protection (infection prevention and reduced infectiousness upon infection". But in fact, these two mechanisms were not formally modeled in this study. The authors should either formally model both parts of the effect (i.e., separately model direct effectiveness and the secondary attack rate) or acknowledge throughout the manuscript that the observed effects are a composite of two effects, and that their relative contribution might differ between individuals (e.g., might be different between previously vaccinated and previously infected).

Response: The reviewer points out that we do not explicitly model indirect protection from the two distinct mechanisms: (1) infection prevention; and (2) reduced infectiousness. We agree that our study design does not distinguish between the two pathways in this study, and that our estimates represent a composite estimate of overall indirect protection. To address this point, we have added clarifications throughout the manuscript to emphasize that we are quantifying a composite measure of indirect protection and include this as a limitation.

In Discussion:

“This study has limitations. **This study design does not distinguish between indirect protection from infection prevention and reduced infectiousness; the relative contribution between these mechanisms may vary between individuals (e.g., immunity status).**”

In Introduction:

“However, few studies have been able to quantify **overall indirect protection, which includes indirect protection due to infection prevention in addition to reduced infectiousness**^{3,11}”

In this study, we use a modified test-negative case-control design to quantify **overall indirect protection from both vaccine-derived and infection-acquired immunity**, including their strength and durability, and considering different vaccine formulations.”

Comment 2

Conclusions from the analysis of bivalent vs. monovalent vaccines (Figure 5 and text) are overstated. First, the two exposures were not directly contrasted (for relative effectiveness), rather both were compared separately to the unvaccinated. Second, estimates of 18-70% vs. 19-38% are at most

"unclear" and certainly do not demonstrate any "trends" (more generally, statistically un-rigorous mentions of "trends" should be removed).

Response: We thank the reviewer for this comment and agree with their suggestion. We recognize we cannot directly compare indirect protection from bivalent and the ancestral monovalent vaccines in this analysis since they were administered at different points during the study period. To address this point, we have removed all language about any "trend" or comparison between the vaccine formulations. Instead, we focus on reporting the key result that the bivalent vaccines provide strong indirect protection within the first three months after vaccination. These changes are outlined below.

In Discussion:

"We also evaluated indirect protection from variant-targeting (bivalent) vaccines and found they provided strong indirect protection ~~and trended toward greater indirect protection compared to the monovalent ancestral vaccine~~ within the first three months of vaccination.

~~**Variant-targeting vaccines, specifically the bivalent vaccine, generate strong indirect protection, with a trend towards greater indirect protection compared to the monovalent ancestral vaccine.."**~~

Comment 3

The authors do well to adjust for the immune status of the case/control. But the immune status of the case/control is scientifically very interesting as a possible effect modifier, and I would be eager to see an analysis of the indirect effect stratified on this immune status. I.e., it is plausible that the indirect effect is much smaller (perhaps entirely non-existent) when the index case is vaccinated. This would also be very relevant to public health policy.

Response: We agree with the reviewer that the immunity status of cases and controls may be an important effect modifier of indirect protection and is highly policy relevant. Of note, nearly 90% of cases and controls received at least one dose of a COVID-19 vaccine (see Table 1), so our primary results reflect indirect protection in a highly vaccinated population. To address this comment, we stratified the study population by the immune status of cases and controls into four groups (no immunity, vaccine-derived immunity only, infection-acquired immunity only, and hybrid immunity) and evaluated indirect protection. The results of these sub-analyses have been added to the supplementary materials as Supplementary Table 10 and are shown below. We have also added descriptions of this analysis to the Methods and Results. Overall, we identify similar strength and durability of indirect protection across all immune status. The exception was infection-acquired immunity only, although the small sample size here was small which somewhat limits interpretation (N=549). Many sub-analyses, especially when estimating durability of indirect protection, are similarly underpowered. Overall, these findings support that indirect protection is similar across immune status of contacts in this high-risk environment within the limitations of this sub-analysis.

In Results (Indirect protection from COVID-19 vaccine and infection-acquired immunity):

"Additionally, both vaccine-derived and infection-acquired indirect protection from roommates was broadly similar among cases and controls with different levels of direct protection (no prior immunity, vaccine-derived immunity only, infection-acquired immunity only, and hybrid immunity), although this analysis was underpowered in some groups (Supplementary Table 10)."

In Methods (Statistical analysis):

“We stratified indirect protection by immune status in cases and controls to explore the relationship between indirect and direct protection.”

In Supplementary Information:

Supplementary Table 10. Indirect protection from vaccine-derived and infection-acquired immunity stratified by immunity of matched cases and controls.

	Odds ratio (OR) (95% CI)				
	Cases and controls with different immunity status				
	All cases and controls (main)	No immunity (N=815)	Only vaccine-derived immunity (N=6,225)	Only infection-acquired immunity (N=543)	Both vaccine- and infection-acquired immunity (hybrid) (N=4,881)
Vaccine-derived immunity					
Any	0.778 (0.694, 0.872)	0.758 (0.551, 1.04)	0.759 (0.643, 0.896)	1.04 (0.666, 1.61)	0.757 (0.619, 0.925)
By dose					
Partially vaccinated	0.925 (0.891, 0.96)	0.899 (0.798, 1.01)	0.918 (0.87, 0.968)	1.02 (0.872, 1.19)	0.926 (0.870, 0.985)
Primary series only	0.856 (0.824, 0.888)	0.808 (0.718, 0.910)	0.842 (0.798, 0.889)	1.04 (0.889, 1.21)	0.857 (0.806, 0.912)
One booster	0.791 (0.763, 0.821)	0.727 (0.645, 0.818)	0.773 (0.732, 0.816)	1.06 (0.906, 1.24)	0.793 (0.746, 0.844)
Two or more boosters	0.732 (0.705, 0.76)	0.653 (0.580, 0.736)	0.71 (0.672, 0.749)	1.08 (0.923, 1.26)	0.734, 0.691, 0.781)
By time					
<3 months	0.703 (0.619, 0.798)	0.704 (0.480, 1.03)	0.686 (0.571, 0.824)	0.964 (0.576, 1.61)	0.672 (0.538, 0.838)
3-6 months	0.871 (0.749, 1.01)	-	0.817 (0.658, 1.01)	-	0.973 (0.757, 1.25)
6-12 months	0.830 (0.727, 0.949)	0.925 (0.608, 1.41)	0.831 (0.684, 1.01)	1.06 (0.617, 1.81)	0.767 (0.611, 0.962)
12+ months	0.820 (0.671, 1.00)	-	0.828 (0.618, 1.11)	-	0.712 (0.514, 0.987)
Infection-acquired immunity					
Any	0.841 (0.773, 0.915)	0.837 (0.583, 1.20)	0.806 (0.715, 0.909)	0.821 (0.558, 1.21)	0.902 (0.788, 1.03)
By time					
<3 months	0.628 (0.451, 0.875)	-	-	-	0.799 (0.512, 1.25)
3-6 months	0.616 (0.476, 0.796)	-	0.702 (0.479, 1.03)	-	0.662 (0.450, 0.973)
6-12 months	0.746 (0.619, 0.898)	-	0.719 (0.540, 0.958)	-	0.724 (0.547, 0.959)
12+ months	0.884 (0.808, 0.967)	0.818 (0.547, 1.22)	0.841 (0.739, 0.956)	0.913 (0.610, 1.37)	0.952 (0.826, 1.1)

We conducted a sub-analysis to evaluate differences in indirect protection based on immunity status in cases and controls. We used separate conditional logistic regression models to quantify indirect protection against Omicron SARS-CoV-2 infection among cases and controls in four groups: no immunity, vaccine-derived immunity only, infection-acquired immunity only, and hybrid immunity. Indirect protection was estimated as 1-OR from the adjusted model. We adjusted for age and risk of severe COVID-19 of both the case/control and their roommate. The sample size was highly variable between sub-groups, limiting inference in some groups. Some analyses were underpowered due to sample size. Results for sub-analyses where sample size was less than 100 cases and controls are not reported.

Comment 4

The authors state "We performed matching of cases and controls to minimize risk of confounding" – but matching in a case-control study does not mitigate confounding (the opposite is true, as usually cited from <https://www.bmj.com/content/352/bmj.i969>, which is also cited in this manuscript). The sentence should be corrected.

Response: We agree with the reviewer and have corrected this sentence. Of note, to address this concern, we performed analyses that controlled for the matched factors in the analysis to ascertain if bias was introduced through matching. This analysis demonstrated stable study results, reducing the likelihood of introduction of bias through matching (Supplementary Table 15).

In Methods:

Statistical analysis

"We performed matching of cases and controls to minimize risk of confounding and to account for differences by building and over time by person-level characteristics and time to improve precision and control for unobserved factors (Supplementary Notes)."

Sensitivity analyses

"Since matching may introduce bias²⁸, we also tested an unconditional logistic regression model with the same study population of matched cases and controls that adjusted for all factors that had been matched exactly, including building and security level, vaccine status of cases and controls, and prior infection status of cases and controls."

In Results (Sensitivity analyses):

"We found similar results when we used unconditional logistic regression to control for all matching factors... (Supplementary Table 15)."

Comment 5

Are the estimates in Figure 3 crude or adjusted? This should be written in the legend.

Response: The estimates in Figure 3 are adjusted for age and risk of severe COVID-19 of both cases and controls and their roommates. We have clarified this point in the legend for Figure 3.

In Table and Figures

"Figure 3. Overall vaccine-derived and infection-acquired indirect protection to close social contacts against SARS-CoV-2 infection.

...We adjusted for age and risk of severe COVID-19 of both the case and control and their roommates."

Comment 6

Perhaps I missed it, but I couldn't find which vaccines were used (other than being monovalent or bivalent) in the study population. mRNA or non-mRNA? Pfizer? Moderna? Etc.

Response: We appreciate the opportunity to clarify. The vast majority of vaccines administered in the population were mRNA vaccines (BNT162b2 and mRNA-1273). Only 17% of vaccinated residents received a Ad26.COVS vaccine as their primary vaccine series (any subsequent booster doses were with mRNA vaccines). Residents that received a Ad26.COVS vaccine were considered fully vaccinated for this analysis. We have added this clarification to the results.

In Results:

“Most residents received mRNA vaccines for their primary series (83%).”

In Methods (Data):

“Isolation, quarantine, and testing practices **and vaccine administration** during the study period are further described in the Supplementary Notes.”

In Supplementary Notes (COVID-19 vaccination):

“COVID-19 vaccination

COVID-19 vaccines became available to residents in the California state prison system in December 2020. Most residents received mRNA vaccines, including BNT162b2 (65%) and mRNA-1273 (18%), as their primary COVID-19 vaccine series. Only 17% of residents received the Ad26.COV2.S vaccine. In our final study population, 6% of cases and controls and 5% of their roommates received the Ad26.COV2.S vaccine. For this analysis, we assumed residents were fully vaccinated after receiving 2 doses of a mRNA vaccine or one dose of the Ad26.COV2.S vaccine.”

Comment 7

The test-negative design mitigates certain biases while potentially causing new ones (<https://pubmed.ncbi.nlm.nih.gov/27587721/>). This should be acknowledged, and the relevant biases discussed.

Response: We agree with the reviewer that the test negative design is subject to potential bias. We have added more discussion on these biases to the Discussion and included the suggested reference.

In Discussion (Limitations):

“This test-negative case-control study design is observational and thus subject to potential confounding²⁷. For example, vaccinated or previously infected residents may have unobserved differences in behavior (e.g., fewer social interactions, a more risk-averse social cohort, masking) that lower their risk of infection and/or transmission to social contacts, and we do not have data on behavior such as masking. However, our analysis demonstrates time-dependent indirect protection (waning) from both sources, which argues against the presence of such confounding factors. The negative control analysis (influenza vaccination) further reduces the likelihood of residual confounding. We also match cases and controls by all observed characteristics (including vaccine and prior infection status), and roommate assignments are expected to be random with respect to vaccine and prior infection status.

While testing is frequent in this population throughout the pandemic, there may be some misclassification of prior infection status or timing due to imperfect case ascertainment²⁷, though any misclassification is more likely nondifferential (Supplementary Table 1).”

In References:

27. Sullivan SG, Tchetgen Tchetgen EJ, Cowling BJ. Theoretical Basis of the Test-Negative Study Design for Assessment of Influenza Vaccine Effectiveness. *Am J Epidemiol.* 2016;184(5):345-353.

Comment 8

The limitation to external generalizability resulting from using data from a prison system should be acknowledged. E.g., it is plausible that the indirect effect is stronger because of the closer cohabitation that occurs in a prison.

Response: We agree that our results are most applicable to high-risk transmission environments with close contacts, such as correctional settings, and reflect indirect protection from roommates in densely populated settings. We have now addressed this limitation to our findings in the Discussion.

In Discussion:

“Our study population is a subset of the overall California incarcerated population during the pandemic that co-resides in close contact with a single person, and our specific estimates on indirect protection are most applicable to similar high-risk transmission environments.

The absolute magnitude of indirect protection against SARS-CoV-2 infection will depend on the transmission environment and social context.”

Comment 9

I would alter the tone of the Discussion section to be less self-congratulatory ("our study makes significant strides", etc.). A modest discussion of each relevant point would be preferable to an enumeration of strengths.

Response: We agree and have edited the text accordingly. We have removed these remarks and streamlined the discussion.

In Discussion:

“...We observe infection-acquired immunity may generate stronger and more durable indirect protection than vaccine-derived indirect protection. This finding becomes most clear when accounting for time since vaccine and/or infection because the study population is more recently vaccinated than infected. The strength of infection-acquired indirect protection could be explained by many mechanisms, including that natural infection generates a more robust immune response. Another explanation is that recent infection occurs with more contemporary variants compared to vaccination with an ancestral strain formulation, meaning infection generates an immune response more tailored to circulating variants. We also find that indirect protection from both vaccine-derived and infection-acquired immunity wanes over time. This could be explained by many immunologic mechanisms (e.g., waning antibody titers), but also by viral evolution and increasing mismatch between the generated host immunity and circulating variants over time...”

Response to Reviewer 2

Remarks to the Author

The authors take advantage of SARS-CoV-2 test data from a surveillance program in California state prisons to determine indirect protection from infection-acquired immunity and its comparative strength and durability to vaccine-derived indirect protection. The study shows that both infection-acquired and vaccine-derived immunity independently yield indirect protection to close social contacts (roommates in a prison) with differences in their strength and waning.

As the authors point out, there is in the current context of high levels of vaccination, prior infection and novel variants of SARS-CoV-2 limited information on overall indirect protection (i.e., the sum of infection prevention and reduced infectiousness upon infection). The value of the study is the fact that this was determined in a well-defined setting using a straight forward and elegant case-control design. A measurement of indirect protection from infection as well as vaccines (including ancestral and variant targeted vaccines) in one study and in a well-defined setting is an original finding, at least to my knowledge.

The study found that infection-acquired immunity generated stronger and more durable indirect protection than vaccine-derived indirect protection. This observation was expected, but the merits of the study is that this was quantified, and the temporal dynamics was explored. On this basis, I expect the findings to be of significance to the field.

Response: We appreciate the reviewer's evaluation of the novelty and significance of our study and for their detailed review and helpful comments.

Comment 1

The study was designed as a test-negative case-control study. Cases were residents with a positive SARS-CoV-2 test, whereas controls were residents with a negative SARS-CoV-2 test. Cases and controls were matched exactly by time (tests within two days), prison building and security level, COVID-19 vaccine status and prior SARS-CoV-2 infection. Cases and controls were then further matched to minimize differences in time since their most recent SARS-CoV-2 infection and/or COVID-19 vaccination as well as age and risk score for severe COVID-19. The analytical strategy and the validity of the results depends on the success of this complicated matching procedure, and the authors are well aware of this. This is reflected among others in the supplemental material which provides insight into the process of matching and also the sensitivity analyses presented in Suppl Table 11. Nonetheless, Table 1 shows minor differences between cases and controls, e.g., 11.5% cases unvaccinated compared with 10.5% controls, and 42% cases with prior infection compared with 44% controls. These differences may be due to the fact that some cases had two controls and others only one, and may not represent a mismatch problem. An explanation will be relevant.

Response: We appreciate the reviewer's attention to the success of the match procedure, and for the opportunity to clarify. Indeed, these differences in the proportion of cases and controls in their vaccine and prior infection status in Table 1 are not because of imperfect matching but because of variable 1:2 matching. These variables were matched exactly. To address this comment about Table 1, we now report weighted statistics on the characteristics of controls and roommates of controls based on match group size in addition to unweighted statistics. We also added clarifying footnotes. These edits are shown below.

Table 1: Characteristics of the study population including COVID-19 cases, matched controls, and their roommates in California prisons.

	Outcome (N (SD) or N (%))			Exposure (N (SD) or N (%))		
	Cases (N=4,640)	Controls ^a		Roommate of cases (N=4,640)	Roommate of controls ^a	
		Unweighted (N=7,824)	Weighted (N=4,640)		Unweighted (N=7,824)	Weighted (N=4,640)
Age (years)	42.3 (12.6)	42.6 (12)	42.3 (11.9)	42.7 (12.4)	42.4 (11.9)	42.2 (11.9)
Sex (male)	4,519 (97)	7,616 (97)	4,518 (97)	4,517 (97)	7,617 (97)	4,518 (97)
Race						
American Indian/Alaskan Native	52 (1)	87 (1)	51 (1)	52 (1)	78 (1)	47 (1)
Asian or Pacific Islander	52 (1)	118 (2)	71 (2)	49 (1)	92 (1)	54 (1)
Black	1,285 (28)	2,084 (27)	1,258 (27)	1,272 (27)	2,088 (27)	1,258 (27)
Hispanic	2,321 (50)	3,988 (51)	2,346 (51)	2,324 (50)	4,016 (51)	2,374 (51)
White	738 (16)	1,179 (15)	704 (15)	733 (16)	1,197 (15)	700 (15)
Other	192 (4)	368 (5)	210 (5)	210 (5)	353 (5)	208 (4)
Risk score for severe COVID-19^b	1.4 (2)	1.3 (1.8)	1.3 (1.8)	1.4 (2)	1.3 (1.8)	1.3 (1.8)
Security level^c						
Low	2,218 (48)	3,907 (50)	2,218 (48)	2,218 (48)	3,907 (50)	2,218 (48)
Moderate	899 (19)	1,455 (18)	899 (19)	899 (19)	1,455 (18)	899 (19)
High	1,523 (33)	2,553 (32)	1,523 (33)	1,523 (33)	2,553 (32)	1,523 (33)
COVID-19 vaccine status						
Unvaccinated	541 (12)	817 (10)	541 (12)	652 (14)	897 (11)	545 (12)
Partially vaccinated	22 (0)	27 (0)	22 (0)	48 (1)	113 (1)	70 (2)
Primary series only	960 (21)	1,503 (19)	960 (21)	979 (21)	1,648 (21)	1,004 (22)
1 booster dose	2,773 (60)	4,881 (62)	2,773 (60)	2,579 (56)	4,533 (58)	2,636 (57)
2+ booster doses	344 (7)	596 (8)	344 (7)	382 (8)	633 (8)	385 (8)
Time since last vaccine dose (days)	155 (122)	147 (120)	150 (122)	151 (125)	148 (128)	149 (129)
Prior SARS-CoV-2 infection status						
Has prior infection	1,953 (42)	3,471 (44)	1,953 (42)	2,181 (47)	4,024 (51)	2,344 (51)
Time since last infection (days)	486 (147)	473 (152)	469 (153)	469 (169)	455 (175)	453 (174)

In this analysis, we estimated indirect protection against SARS-CoV-2 infection from differences in the COVID-19 vaccine status and prior SARS-CoV-2 infection status of among roommates of matched cases and controls. **Cases were matched with controls in a 1:2 ratio, although a subset were matched 1:1 based on available controls meeting study criteria. Cases and controls were matched exactly by vaccine status, prior infection status, building, and security level, with additional distance matching on other variables.** Full matching process is described in the Methods.

^aDifferences in estimates for variables matched exactly between cases and unweighted controls (i.e., vaccine status, prior infection status, and security level) are due to variable 1:2 matching. We report weighted estimates of these variables to account for different match group size.

^bRisk for severe COVID-19 is a composite score of the number of risk factors for severe disease including age and medical conditions. Risk scores were calculated by the California Correctional Health Care Services.

^cResident contact and activity are determined by their security level. Low level (score 1-2) reflects the lowest security level with more social contacts, while high level (score 4) is the highest security level with fewer social contacts.

Comment 2

The authors mention that 2179 residents were included more than once, but the observations were treated as independent. I'm not a statistician, but I would think that repeated observations should be taken into account, at least by providing robust confidence intervals (the intervals presented may be too narrow, but the estimates are likely to be valid). Most software packages will provide solutions for this. This issue should be addressed before the paper can be accepted.

Response: We appreciate the reviewer's comment about accounting for repeated measures in residents within the study population. We estimate indirect protection with a conditional logistic regression model, so indirect protection is estimated within matched groups. Even though some residents are included as cases and controls throughout the study, residents are not repeated within a matched group. However, we agree with the reviewer that it is important to ensure that our results are not impacted by repeated measures when residents are included in a later matched group. To address this comment, we perform two analyses. First, we repeated the analysis using robust standard errors in a sensitivity analysis with an unconditional logistic regression model. In the model, we adjust for all matching factors and generate person-level cluster robust confidence intervals. Second, we conducted an analysis without repeated measures (residents were only eligible for inclusion once during the study period). Our results and interpretation for both analyses remain consistent, and we include additions to the text below.

In Results (Sensitivity analyses):

“We found similar results when we ... accounted for repeated observations of residents in the study population over time (Supplementary Table 15).”

In Methods:

Statistical analysis

“We did not allow residents to be repeated within a matched group... We treated repeated observations of a single resident in different matched groups independently, although we tested this assumption in sensitivity analyses.”

Sensitivity analyses

“To assess the sensitivity of our results to repeated measures of the same residents, we fit a conditional logistic regression model without repeated measures and an unconditional logistic regression model with person-level cluster robust errors.”

In Supplementary Information:

Supplementary Notes (Additional notes on statistical analysis and data)

“Residents were eligible for inclusion throughout the study period based on their testing data; however, residents were not included more than once within a matched group. Of the 9,625 residents included in the study, most were included only once as a case or control (N=7,555), and 2,070 residents were included more than once as a case and/or control in distinct matched groups. Each observation of a resident was treated independently in the main analysis since only a minority of individuals were repeated and the within person correlation may be less significant in this analysis. However, to evaluate this assumption, we assessed the sensitivity of our results to repeated measures in multiple sensitivity analyses (Supplementary Table 15), including removal of repeated residents.”

Supplementary Table 15. Sensitivity analyses of regression model specifications for indirect protection from vaccine and infection-acquired immunity.

	Odds ratio (OR) (95% CI)			
	Conditional logistic regression (main)	Unconditional logistic regression	Conditional logistic regression without repeated measures	Unconditional logistic regression with person-level clustering
Vaccine-derived immunity				
Any	0.778 (0.694, 0.872)	0.800 (0.715, 0.896)	0.799 (0.697, 0.916)	0.800 (0.709, 0.904)
By dose				
Partially vaccinated	0.925 (0.891, 0.96)	0.938 (0.904, 0.973)	0.934 (0.893, 0.976)	0.938 (0.902, 0.975)
Primary series only	0.856 (0.824, 0.888)	0.879 (0.848, 0.912)	0.872 (0.833, 0.912)	0.879 (0.845, 0.915)
One booster	0.791 (0.763, 0.821)	0.825 (0.795, 0.855)	0.814 (0.778, 0.851)	0.825 (0.793, 0.858)
Two or more boosters	0.732 (0.705, 0.76)	0.773 (0.745, 0.802)	0.76 (0.726, 0.795)	0.773 (0.743, 0.804)
By time				
<3 months	0.703 (0.619, 0.798)	0.735 (0.650, 0.831)	0.743 (0.638, 0.865)	0.735 (0.644, 0.839)
3-6 months	0.871 (0.749, 1.01)	0.878 (0.761, 1.01)	0.834 (0.698, 0.998)	0.878 (0.753, 1.02)
6-12 months	0.830 (0.727, 0.949)	0.851 (0.748, 0.968)	0.856 (0.73, 1.00)	0.851 (0.740, 0.978)
12+ months	0.820 (0.671, 1.00)	0.835 (0.689, 1.01)	0.837 (0.661, 1.06)	0.835 (0.683, 1.02)
Infection-acquired immunity				
Any	0.841 (0.773, 0.915)	0.847 (0.781, 0.918)	0.831 (0.752, 0.919)	0.847 (0.776, 0.925)
By time				
<3 months	0.628 (0.451, 0.875)	0.665 (0.481, 0.907)	0.556 (0.368, 0.84)	0.665 (0.479, 0.921)
3-6 months	0.616 (0.476, 0.796)	0.667 (0.525, 0.843)	0.535 (0.382, 0.748)	0.667 (0.517, 0.861)
6-12 months	0.746 (0.619, 0.898)	0.781 (0.658, 0.925)	0.812 (0.645, 1.02)	0.781 (0.652, 0.936)
12+ months	0.884 (0.808, 0.967)	0.884 (0.811, 0.963)	0.870 (0.782, 0.968)	0.884 (0.805, 0.970)

In our primary analysis, we used a conditional logistic regression model to assess indirect protection within groups of matched cases and controls. Here we assessed the sensitivity of results to potential bias from matching by fitting an unconditional logistic regression model with additional adjustment for factors that were exactly matched between cases and controls (vaccine and prior infection status of cases and controls and building and security level). To evaluate robustness of results to repeated observations from cases and controls, we fit a conditional logistic regression model without repeated measures and fit an unconditional logistic regression model with person-level cluster robust errors.

Comment 3

A minor issue is the headings of Table 1. Many readers will look at this table before reading the methods section. The roommates reflect the exposure status of cases/controls. The table may be misunderstood in a way that the roommates represent a second set of cases/controls. Perhaps a slight edition will make it easier for the uninitiated reader to understand the design and the table.

Response: We have made changes to Table 1 that clarify the difference between cases and controls and the role of roommates in the study design. We now label the “Outcome” (Case/control) and “Exposure” (Roommate of case/control) in the modified Table 1. We further clarify these differences in the table footnote.

Comment 4

The supplementary material presents the results of a logistic regression analysis (column 6 in Supplementary Table 11). It is unconditional I assume - should perhaps be spelled out.

Response: The logistic regression analysis in this sensitivity analysis is unconditional. We have added clarifications to the main text and to Supplementary Table 15.

In Results (Sensitivity analyses):

“We found similar results when we used **unconditional logistic regression** to control for all matching factors... (Supplementary Table 15).”

In Supplementary Information:

Supplementary Table 15. Sensitivity analyses of regression model specifications for indirect protection from vaccine and infection-acquired immunity.

	Odds ratio (OR) (95% CI)		
	Conditional logistic regression (main)	Unconditional logistic regression	...
Vaccine-derived immunity			
Any	0.778 (0.694, 0.872)	0.800 (0.715, 0.896)	...
By dose			
Partially vaccinated	0.925 (0.891, 0.96)	0.938 (0.904, 0.973)	...
Primary series only	0.856 (0.824, 0.888)	0.879 (0.848, 0.912)	...
One booster	0.791 (0.763, 0.821)	0.825 (0.795, 0.855)	...
Two or more boosters	0.732 (0.705, 0.76)	0.773 (0.745, 0.802)	...
By time			
<3 months	0.703 (0.619, 0.798)	0.735 (0.650, 0.831)	...
3-6 months	0.871 (0.749, 1.01)	0.878 (0.761, 1.01)	...
6-12 months	0.830 (0.727, 0.949)	0.851 (0.748, 0.968)	...
12+ months	0.820 (0.671, 1.00)	0.835 (0.689, 1.01)	...
Infection-acquired immunity			
Any	0.841 (0.773, 0.915)	0.847 (0.781, 0.918)	...
By time			
<3 months	0.628 (0.451, 0.875)	0.665 (0.481, 0.907)	...
3-6 months	0.616 (0.476, 0.796)	0.667 (0.525, 0.843)	...
6-12 months	0.746 (0.619, 0.898)	0.781 (0.658, 0.925)	...
12+ months	0.884 (0.808, 0.967)	0.884 (0.811, 0.963)	...

In our primary analysis, we used a conditional logistic regression model to assess indirect protection within groups of matched cases and controls. **Here we assessed the sensitivity of results to potential bias from matching by fitting an unconditional logistic regression model with additional adjustment for factors that were exactly matched between cases and controls (vaccine and prior infection status of cases and controls and building and security level)...**